# Diagnosis of Carnitine Deficiency in Extremely Preterm Neonates Related to Parenteral Nutrition: Two Step Newborn Screening Approach

**DOI:** 10.3390/ijns5030029

**Published:** 2019-08-31

**Authors:** Mamatha Ramaswamy, Victor Anthony Skrinska, Rola Fayez Mitri, Ghassan Abdoh

**Affiliations:** 1Metabolic Laboratory, Department of Pathology and Laboratory Medicine, Hamad Medical Corporation, Doha PO box 3050, Qatar; 2Newborn Screening Unit, Hamad Medical Corporation, Doha PO box 3050, Qatar

**Keywords:** carnitine, preterm neonates, premature infants, parenteral nutrition, TPN

## Abstract

Currently, there is no evidence in the literature to support the routine supplementation of all parenterally fed premature infants with l-carnitine. In our study, we found that about 8.56% of extremely preterm neonates are diagnosed with carnitine deficiency secondary to malnutrition, either due to reduced stores at birth or related to total parenteral nutrition (TPN). Our two step approach of performing newborn screening (NBS) again at 32 weeks gestational age (GA) equivalent helps to diagnose 81.4% more preterm babies with carnitine deficiency—who would otherwise be missed—and supplement them with l-carnitine for optimal growth. We performed a retrospective cohort study to diagnose carnitine deficiency related to malnutrition in two groups: those presenting at birth and those presenting later in life. We found that there was a statistically significant difference in the median GA and birth weight (BW) between the two groups, but there was no difference in the free carnitine levels.

## 1. Introduction

Carnitine is an amino acid that plays an important role in the transport of long chain fatty acids into mitochondria where β oxidation occurs [1]. It also can fulfil a detoxifying role by forming and exporting acylcarnitine esters of acyl-CoA molecules that accumulate within the mitochondria. Plasma and tissue concentrations of carnitine are low in newborn infants compared with those found in older children [2]. This may be due to the immaturity of synthetic pathways [3] or reflect a lower renal threshold [4]. Both infant formulas and breast milk contain carnitine [5], but carnitine is absent in most parenteral feeding formulations. In term infants, a major deficiency of carnitine is unlikely provided oral feeding is established. Preterm infants may be at risk of deficiency because of the immaturity of synthetic pathways and renal mechanisms together with delayed oral feeding [6]. The l-carnitine reserves in a full term newborn are approximately 25–50% of those in adults [2], and the reserves in preterm neonates are even lower [7].

Recent studies published in the literature do not support prophylactic carnitine supplementation in parenteral nutrition for preterm infants. Premature infants of less than 34 weeks gestation and requiring total parenteral nutrition develop nutritional carnitine deficiency with impaired fatty acid oxidation and ketogenesis, and carnitine supplementation improves this metabolic disturbance [8]. The aim of the current study is to diagnose carnitine deficiency in extremely preterm neonates (born before 32 weeks gestational age) through newborn screening and then supplement them with l-carnitine in parenteral feeding for optimal growth. l-carnitine is not routinely provided in parenteral nutrition solutions in Qatar. Extremely premature neonates have a reduced capacity to synthesize carnitine and may need supplementation if they are on long term total parenteral nutrition (TPN). Placental transport of carnitine to the fetus occurs primarily during third trimester of pregnancy [9]. Carnitine deficiency develops among very low birth weight infants who do not receive exogenous supplementation for two to four weeks [10]. This is related to minimal body stores at birth, limited carnitine production, high carnitine loss in the urine, and lack of L-carnitine supplements in standard parenteral solutions [7,9]. Newborn screening (NBS) in Qatar is performed on dried blood spots (DBS) collected at or after 36 hours of birth. For extremely preterm infants born before 32 weeks gestational age (GA), our practice is to collect a successive DBS at 32 weeks GA equivalent (S32 GA). This practice helps to identify metabolic disorders that may be missed on the initial NBS due to prematurity and also to diagnose carnitine deficiency related to TPN.

## 2. Materials and Methods

Carnitine deficiency (free carnitine <8.0 μmol/L) was diagnosed in extremely preterm infants on newborn screening by analysis of amino acid/acylcarnitine (AA/AC) profiles using liquid chromatography tandem mass spectrometry. We used the same cut-off for free carnitine as with term infants as there is no well-established cut-off for diagnosis of carnitine deficiency in preterm infants. Other acylcarnitines were also low in these neonates, particularly acetyl-carnitine (C2), propionyl-carnitine (C3), l-palmitoyl-carnitine (C16), stearoyl-carnitine (C18), elaidic-carnitine (C18;1), and the Acyl/Cit ratio. Analysis of AA/AC profiles was done on initial DBS followed by a successive 32 weeks GA DBS on all extremely preterm infants. We also retrospectively collected data on birth weight (BW), gestational age (GA) at birth, and free carnitine levels in infants who were diagnosed with carnitine deficiency for a period of three years (2016–2018). We grouped infants into two cohorts. The first cohort included neonates who were diagnosed with carnitine deficiency on initial DBS. The second cohort included infants whose carnitine levels were normal on initial DBS but diagnosed later with carnitine deficiency on S32 GA equivalent DBS, which was related to TPN. We used statistical analysis Mann Whitney U test to see if there is a statistically significant difference in BW, GA, and free carnitine (C0) levels between the two cohorts when diagnosed with carnitine deficiency.

Once the diagnosis of carnitine deficiency was made, all preterm infants were supplemented with l-carnitine in the TPN. Neonates in both the cohorts were subsequently followed up by another DBS screen for the AA/AC profile, after TPN was stopped, to ensure C0 was normalized before discharge from the hospital. Hence, all babies with primary carnitine uptake defect were excluded from this study, and carnitine deficiency was secondary to malnutrition, either due to reduced body stores at birth in the first cohort or related to TPN in the second cohort. 

## 3. Results

We screened a total of 84,441 newborn babies from 2016 and 2018. During this period, 864 (1.023%) extremely premature infants were also screened. Only preterm infants born before 32 weeks GA were included in this study. Seventy-four extremely premature (8.56%) neonates were diagnosed with carnitine deficiency. Of these, only 13 babies (1.50%) were diagnosed with carnitine deficiency on the initial screen, whereas 61 babies (7.06%) were diagnosed later in the subsequent S32 week GA equivalent screen. The second cohort included four infants diagnosed with carnitine deficiency at GA 31 weeks, and initial DBS was received in these infants at around 32 weeks GA. We have excluded this number from some of the statistics.

The GA for the first cohort ranged from 23 to 29 weeks (median 24 weeks), and for the second cohort 23 to 31 weeks (median 28 weeks). The BW for the first cohort ranged from 590 to 1090 g (median 790 g), and for the second cohort 610 to 1890 g (median 920 g). The C0 levels in the first cohort ranged from 3.56 to 7.69 μmol/L (median of 5.95 μmol/L) and in the second cohort from 3.06 to 7.54 μmol/L (median 5.49 μmol/L). We used non-parametric methods to compare and test whether there is a significant statistical difference in the median GA, BW, and C0 levels between the two cohorts. We used the Mann–Whitney U test calculator from the Social Science Statistics website [11]. Using a two-tailed hypothesis, the U-value obtained from the calculator for GA and BW were less than the expected U-value from the critical value table of the Mann–Whitney U (α = 0.05). However, the U-value for free carnitine was higher than the expected U-value from the table. We found that there was a statistically significant difference in the median GA at birth (Z score = 2.559, *p* = 0.00523) and median BW (Z score = 2.496, *p* = 0.00621) between the two cohorts. However, there was no statistically significant difference in the median C0 levels (Z score = 0.5167, *p* = 0.30153) between the two groups.

The results are displayed in Table 1.

## 4. Discussion

In our study, only 8.56% of the extremely preterm neonates developed carnitine deficiency. Since this percentage is relatively small, prophylactic L-carnitine supplementation in TPN in all extremely preterm infants is not clinically justified. Current evidence in the literature also does not support this clinical practice. Recent systematic review shows that routine supplementation of carnitine in parenteral nutrition of preterm newborns may help to increase carnitine levels, but neither a relevant improvement in the lipid profile nor an increase in weight gain or a decrease in morbimortality or reduction in hospital stay was seen [12]. This study shows that 91.44% of the extremely premature infants do not develop secondary carnitine deficiency related to malnutrition.

In our study, we discovered that only 13 out of 70 preterm infants (18.58%) were diagnosed with carnitine deficiency on the first NBS. The majority, which was 57 out of 70 babies (81.42%), were actually diagnosed with carnitine deficiency on the second newborn screen. The first screen in all these extremely preterm infants showed normal free carnitine levels. The carnitine deficiency in these infants was secondary to parenteral nutrition and would have been missed if the repeat NBS was not performed. Primary carnitine uptake defect was excluded in all these neonates. This study is the first of its kind in literature using a two-step newborn screening strategy for the diagnosis of carnitine deficiency related to TPN in extremely preterm infants. The results of this study clearly shows that NBS programs not following a two-step procedure are likely to be missing the diagnosis of secondary carnitine deficiency related to TPN in their extremely preterm infants. Repeating the AA/AC profile at 32 weeks GA equivalent or a similar practice helps to diagnose carnitine deficiency in extremely preterm infants who are receiving TPN for a while, and timely intervention with L-carnitine supplements can be provided. Our study showed that only 1.02% of the NBS were from extremely premature infants, making it possible for other newborn screening programs to adapt this practice. This kind of approach requires collaboration between the laboratory and clinical teams involved in the diagnosis and management of these tiny patients. Since all these extremely preterm infants are hospitalized, it is easy to obtain a successive DBS for a second screen.

Extremely premature babies have poor sucking reflex, hence they are likely to be dependent on TPN for their complete nutritional needs especially during period of infections and illness. Our study shows that 8.56% of them develop secondary carnitine deficiency related to malnutrition (either due to reduced stores or due to TPN) and will need l-carnitine supplementation. Preterm newborns on total parenteral nutrition have both a reduction in carnitine intake and in tissue reserves. Given that they tend to be more demanding due to their rapid growth, it is not surprising that newborns fed with TPN without supplements will reach very low carnitine levels after two weeks of life [13]. The lack of carnitine can be an etiological factor in the limited ability of preterm newborns to use parenteral lipids. In vitro studies have suggested that fatty acid oxidation will be irregular when levels of tissue carnitine are below 10% of their normal level [14]. 

Our cohort study included an unequal number of samples in each cohort and we were unsure about the normality of the distribution of the data in both groups, hence we used a non-parametric method of statistical analysis. While performing the Mann–Whitney U test, to avoid tied ranks, we used days plus weeks at birth for GA, single digits in birth weight, and decimal places in carnitine levels. There was a statistically significant difference in the gestational age and birth weight between the two groups. The first cohort of preterm infants presented with carnitine deficiency on first NBS and needed supplementation with L-carnitine in TPN from the beginning of their lives. The second cohort of premature neonates were diagnosed with a deficiency only on the second NBS. There was no statistically significant difference in the free carnitine levels between the two cohorts at the time of diagnosis of deficiency. The cohort study presented here was performed on a small number of infants and is in need of more research in the future. The results obtained are very similar to what has already published and emphasizes that extremely preterm infants born with lower birth weight and lower GA have low carnitine reserves at birth and are more likely to present with carnitine deficiency earlier than later. Thus, these infants need l-carnitine supplementation in TPN from the beginning of their lives. However, the severity of carnitine deficiency has no relation to GA or BW. The findings from this study may not be relevant to preterm infants born after successive 32 weeks GA.

## 5. Conclusions

The majority (>90%) of the extremely preterm infants do not develop secondary carnitine deficiency related to malnutrition. Hence, there is no role for prophylactic l-carnitine supplementation in TPN. The majority (>80%) of the extremely preterm neonates who develop carnitine deficiency do so after being on TPN for a while, and will benefit from the two-step newborn screening approach for a timely diagnosis of carnitine deficiency and treatment with l-carnitine supplements in TPN.

## Figures and Tables

**Table 1 IJNS-05-00029-t001:** Carnitine deficiency cases diagnosed on newborn screening (NBS) during a three-year period.

	2016	2017	2018	Total
Diagnosis of secondary carnitine deficiency on initial DBS	8	2	3	13
Diagnosis of secondary carnitine deficiency on S32 week GA DBS	13	33	15	61
Total cases of carnitine deficiency	21	35	18	74
Total number of extremely preterm babies	263	311	290	864
Total number of newborn screens	27,188	28,608	28,645	84,441

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
