# Peer review of "Diagnosis of Carnitine Deficiency in Extremely Preterm Neonates Related to Parenteral Nutrition: Two Step Newborn Screening Approach"

_2409-515X, 2019, doi:10.3390/ijns5030029_

Round 1

Reviewer 1 Report

I consider that this review addresses a subject of outstanding interest in the field of Neonatology. It is a well structured work, easy to read and understandable. Nevertheless, I believe that some aspects, described below, should be taken into account in the review to adequately assess the risk of carnitine deficiency in extremely premature infants.

1- I should thank if the authors indicate why they establish the limit to consider carnitine deficiency in free carnitine levels of 8.0umol / L, since the cutoff point in premature newborns is not well established.

2- In the material and methods section I consider that the type of food received should be included since it is a factor that clearly influences the contribution of carnitine received. Carnitine content varies, being higher in formula milk than in breastfeeding. It has been shown that preterms fed with preterm formulas maintain higher levels of plasma carnitine than those fed with fortified breast milk. Therefore, it would be important to know the days of exclusive parenteral feeding, days of life at the beginning of trophic feeding and the type of oral feeding received; and correlate these aspects with the risk of evolutionary carnitine deficiency.

3- In the results section the authors show the median weight of the two cohorts evaluated, but in the abstract the median weight of the first cohort (774 grams) is extrapolated as the cut-off point below which the risk of carnitine deficiency due to malnutrition at birth is greater, however, I consider that this point is not exact and deserves a more rigorous analysis.

4- Although there is a discrepancy in this regard in the literature, an important factor to consider in the risk of carnitine deficiency is intrauterine growth retardation. It would be of interest to specify the percetange of premature infants with low weight for gestational age and with intrauterine growth retardation in both cohorts.

5- As indicated in the introduction, in preterm infants of gestational age less than 34 weeks it is convenient to repeat the sample for neonatal screening as the immaturity in preterm newborns of different systems, such as the hypothalamic-pituitary-thyroid axis; or hepatic or renal enzymatic immaturity; can affect its results. Current international recommendations advocate repeating a second determination between the second and fourth week of life in order to minimize the influence of these modifying factors on the effectiveness of screening. I would be grateful if you could clarify whether the practice of the second sample at 32 weeks of corrected age referred to in the article is the accepted practice of screening in Qatar or in your hospital in particular, because although it may be useful for diagnosing evolutionary carnitine deficiency in preterm newborns of gestational age close to 32 weeks, it may difficult the diagnose of other entities.

Author Response

Many thanks to both reviewers for their detailed and constructive feedback to improve the article. I have provided the responses for each point in blue sentences and changes made in the article in red sentences.

Reviewer 1 comments

I consider that this review addresses a subject of outstanding interest in the field of Neonatology. It is a well structured work, easy to read and understandable. Nevertheless, I believe that some aspects, described below, should be taken into account in the review to adequately assess the risk of carnitine deficiency in extremely premature infants.

1- I should thank if the authors indicate why they establish the limit to consider carnitine deficiency in free carnitine levels of 8.0umol / L, since the cutoff point in premature newborns is not well established.

Thanks for making this comment. We have included a sentence in the methods as follows

We used the same cut off for free carnitine as with term infants as there is no well established cut-off for diagnosis of carnitine deficiency in preterm infants.

2- In the material and methods section I consider that the type of food received should be included since it is a factor that clearly influences the contribution of carnitine received. Carnitine content varies, being higher in formula milk than in breastfeeding. It has been shown that preterms fed with preterm formulas maintain higher levels of plasma carnitine than those fed with fortified breast milk. Therefore, it would be important to know the days of exclusive parenteral feeding, days of life at the beginning of trophic feeding and the type of oral feeding received; and correlate these aspects with the risk of evolutionary carnitine deficiency.

Since this was a retrospective study unfortunately we do not have this information about how much formula, breast milk or TPN was given to each infant. But this is a very good point and we will gather this data for our future studies.

3- In the results section the authors show the median weight of the two cohorts evaluated, but in the abstract the median weight of the first cohort (774 grams) is extrapolated as the cut-off point below which the risk of carnitine deficiency due to malnutrition at birth is greater, however, I consider that this point is not exact and deserves a more rigorous analysis.

Sure we agree with you and removed this sentence from the abstract. Following is the modified abstract.

Abstract: Currently there is no evidence in the literature to support the routine supplementation of all parenterally fed premature infants with L-carnitine. In our study we found that about 8.56% of the extremely preterm neonates are diagnosed with carnitine deficiency secondary to malnutrition, either due to reduced stores at birth or related to total parenteral nutrition (TPN). Our two step approach of performing newborn screening (NBS) again at 32 weeks gestational age (GA) equivalent helps to diagnose 81.4% more of the preterm babies with carnitine deficiency which would otherwise be missed and supplement with L-carnitine for optimal growth. We performed a retrospective cohort study to diagnose carnitine deficiency related to malnutrition in two groups, those presenting at birth and those presenting later in life. We found that there was a statistically significant difference in the median GA and birth weight (BW) between the two groups, but there was no difference in the free carnitine levels.

4- Although there is a discrepancy in this regard in the literature, an important factor to consider in the risk of carnitine deficiency is intrauterine growth retardation. It would be of interest to specify the percetange of premature infants with low weight for gestational age and with intrauterine growth retardation in both cohorts.

Sorry we currently do not have this information on our blood cards. But this is a very good point and we will gather this data for our future studies.

5- As indicated in the introduction, in preterm infants of gestational age less than 34 weeks it is convenient to repeat the sample for neonatal screening as the immaturity in preterm newborns of different systems, such as the hypothalamic-pituitary-thyroid axis; or hepatic or renal enzymatic immaturity; can affect its results. Current international recommendations advocate repeating a second determination between the second and fourth week of life in order to minimize the influence of these modifying factors on the effectiveness of screening. I would be grateful if you could clarify whether the practice of the second sample at 32 weeks of corrected age referred to in the article is the accepted practice of screening in Qatar or in your hospital in particular, because although it may be useful for diagnosing evolutionary carnitine deficiency in preterm newborns of gestational age close to 32 weeks, it may difficult the diagnose of other entities.

Its true that our current practice in Qatar is to collect a second DBS at 32 weeks GA equivalent. Our newborn screening covers private hospitals and clinics and for ease of collection and practicalities of not missing the second screen, we have standardized second collection at 32 wks GA. I agree that current international recommendations advocate repeating a second determination between the second and fourth week of life in order to minimize the influence of these modifying factors on the effectiveness of screening. We have to work collaboratively with all stakeholders to change this practice in future.

Reviewer 2 Report

Method/Statistics

The authors should describe the software used for the test.

Were the measured values normally distributed ?

Non-parametric tests are more appropriate when the distribution is not normal.

Authors grouped patients into two cohorts in this study. I think that the data of two cohorts did not correspond. When comparing between groups, "paired t-test" is not appropriate.

Author Response

Many thanks to both reviewers for their detailed and constructive feedback to improve the article. I have provided the responses for each point in blue sentences and changes made in the article in red sentences.

Reviewer  2 comments

Method/Statistics

The authors should describe the software used for the test.

Were the measured values normally distributed ?

Non-parametric tests are more appropriate when the distribution is not normal.

Authors grouped patients into two cohorts in this study. I think that the data of two cohorts did not correspond. When comparing between groups, "paired t-test" is not appropriate.

Thanks for the above comments. When sample number is more than 10 it is said that parametric method is acceptable, hence we used t-test. But we agree with you the number of samples in the two groups don’t correspond. Hence we have replaced the statistics with Mann Whitney U test. We have included reference for the statistics used. We have made the following changes.

Results

The GA for the first cohort ranged from 23 wks to 29 wks (median 24 wks) and for second cohort 23 wks to 31 wks (median 28 wks). The BW for the first cohort ranged from 590 gms to 1090 gms (median 790 gms) and for second cohort 610 gms to 1890 gms (median 1710 gms). The C0 levels in first cohort ranged from 3.56 – 7.69 μmol/L (median of 5.95 μmol/L) and in the second cohort from 3.06 – 7.54 μmol/L (median 5.49 μmol/L). We used non parametric method to compare and test whether there is a statistical significant difference in the median GA, BW and C0 levels between the two cohorts. We used Mann-Whitney U test calculator from the Social Science Statistics website [11]. Using a two-tailed hypothesis, the U-value obtained from the calculator for GA and BW were less than the expected U-value from the critical value table of the Mann-Whitney U (α 0.05). However the U-value for free carnitine was higher than the expected U-value from the table. We found that there was a statistically significant difference in the median GA at birth (Z score – 2.559, p - 0.00523) and median BW (Z score – 2.496, p - 0.00621) between the two cohorts. However there was no statistically significant difference in the median C0 levels (Z score – 0.5167, p – 0.30153) between the two groups.

Discussion

Our cohort study included unequal number of samples in each cohort and we are unsure about the normality of the distribution of the data in both groups, hence we used non parametric method of statistical analysis. While performing Mann Whitney U test, to avoid tied ranks we used days plus weeks at birth for GA, single digits in birth weight and decimal places in carnitine levels. There was a statistically significant difference in the gestational age and birth weight between the two groups. The first cohort of preterm infants presented with carnitine deficiency on first NBS and needed supplementation with L-carnitine in TPN from the beginning of their lives. The second cohort of premature neonates were diagnosed with a deficiency only on the second NBS. There was no statistically significant difference in the free carnitine levels between the two cohorts at the time of diagnosis of deficiency. The cohort study presented here was performed on small number of infants and needs more research in the future. The results obtained are very similar to what is already published and emphasizes that extremely preterm infants born with lesser birth weight and lower GA have low carnitine reserves at birth and are more likely to present with carnitine deficiency earlier than later, and need L-carnitine supplementation in TPN from the beginning of life. However the severity of carnitine deficiency has no relation to the GA or BW. The findings in this study may not be relevant to preterm infants born after successive 32 week GA.

References

https://www.socscistatistics.com/tests/mannwhitney/